# Influenza A virus is transmissible via aerosolized fomites

Sima Asadi [1], Nassima Gaaloul ben Hnia [2], Ramya S. Barre [2,8], Anthony S. Wexler [3,4,5,6], William D. Ristenpart [1] & Nicole M. Bouvier [2,7✉]

Influenza viruses are presumed, but not conclusively known, to spread among humans by several possible routes. We provide evidence of a mode of transmission seldom considered for influenza: airborne virus transport on microscopic particles called "aerosolized fomites." In the guinea pig model of influenza virus transmission, we show that the airborne particulates produced by infected animals are mainly non-respiratory in origin. Surprisingly, we find that an uninfected, virus-immune guinea pig whose body is contaminated with influenza virus can transmit the virus through the air to a susceptible partner in a separate cage. We further demonstrate that aerosolized fomites can be generated from inanimate objects, such as by manually rubbing a paper tissue contaminated with influenza virus. Our data suggest that aerosolized fomites may contribute to influenza virus transmission in animal models of human influenza, if not among humans themselves, with important but understudied implications for public health.

[1] Department of Chemical Engineering, University of California Davis, One Shields Ave., Davis, CA 95616, USA. [2] Department of Microbiology, Icahn School of Medicine at Mount Sinai, 1 Gustave L. Levy Place, New York, NY 10029, USA. [3] Department of Mechanical and Aerospace Engineering, University of California Davis, One Shields Ave., Davis, CA 95616, USA. [4] Air Quality Research Center, University of California Davis, One Shields Ave., Davis, CA 95616, USA. [5] Department of Civil and Environmental Engineering, University of California Davis, One Shields Ave., Davis, CA 95616, USA. [6] Department of Land, Air and Water Resources, University of California Davis, One Shields Ave., Davis, CA 95616, USA. [7] Division of Infectious Diseases, Department of Medicine, Icahn School of Medicine at Mount Sinai, 1 Gustave L. Levy Place, New York, NY 10029, USA. [8]Present address: Department of Ecology and Evolutionary Biology, 304 Guyot Hall, Princeton University, Princeton, NJ 08544, USA. ✉email: nicole.bouvier@mssm.edu

Seasonal epidemic influenza causes hundreds of thousands of deaths worldwide every year[1], punctuated by occasional pandemics with death tolls reaching into the millions[2,3]. Precisely how influenza spreads among humans has long been a matter of debate[4–11]; however, there is broad agreement about the possible modes of transmission between humans[6–11]. Direct or indirect contact modes require a susceptible person to self-inoculate by, for instance, touching one's nose with a virus-contaminated hand; "direct" indicates that person-to-person contact transfers the virus between infected and susceptible hosts, whereas "indirect" implies transmission via a fomite, which is an object like a doorknob or toy that has been contaminated with infectious virus[9]. Airborne transmission may occur by two modes, either by sprays of virus-laden respiratory droplets, such as from a cough or sneeze, impacting immediately onto the respiratory mucosa of a susceptible individual, or by the eventual inhalation of droplet nuclei, microscopic aerosol particles consisting of the residual solid cores of evaporated respiratory droplets[9]. The relative contribution of each of these transmission modes remains unknown, and viral, host, or environmental factors may affect which modes are favored in different settings[11–13].

Uncertainty surrounding the modes by which influenza virus transmits among humans under different conditions hinders the assessment of non-pharmaceutical interventions designed to prevent influenza's spread[14,15]. Animal models of influenza virus transmission are often used to try to elucidate these uncertainties, and to test the efficacy of vaccines[13], under controlled laboratory conditions. Typically, in modeling airborne influenza virus transmission, a virus-donor animal, inoculated with an influenza virus, and a virus-recipient animal, naive to influenza virus, are physically separated in cages that share a common air space, such that infectious particles generated by the donor animal must travel through the air to infect the susceptible recipient[16]. However, the physical nature of these infectious particles is just beginning to be elucidated[17–22]; thus, in this work we seek to characterize the airborne particles that pass between inoculated donor and susceptible recipient in the guinea pig model of influenza virus transmission. We first show that the vast majority of airborne particulates emitted from a guinea pig cage are non-respiratory in origin and thus presumably environmental dust. We then demonstrate that infected guinea pigs heavily contaminate their fur and surrounding environment with virus, and we further establish that if these dust particulates become contaminated with influenza virus they can serve as aerosolized fomites that carry the virus to a susceptible guinea pig through the air. Finally, we show that aerosolized fomites can be generated from inanimate objects, such as by rubbing a virus-contaminated paper tissue. We conclude by discussing the implications of aerosolized fomites for respiratory virus transmission in other animal models and between humans.

## Results

### Aerosolization of non-respiratory particles by guinea pigs. To measure the airborne particulates emanating from the cages of uninfected guinea pigs, we sampled air from a HEPA-filtered guinea pig cage with an aerodynamic particle sizer (APS) that enumerates particles in the size range of 0.3–20 μm. A camera placed above the cage simultaneously captured guinea pig movement over time (Fig. 1a; Supplementary Fig. 1a–c). We then measured airborne particle production by uninfected guinea pigs, placed individually, awake and unrestrained, in the cage. We found that airborne particulates were generated primarily as irregular, sharp spikes, up to 1000 particles s$^{-1}$, and observed that the particle count spikes were almost entirely coincident with

guinea pig motion (Fig. 1b; Supplementary Fig. 2a). Plotting the 1-min time-averaged guinea pig movement velocity, $\bar{V}_{(1)}$, with the corresponding time-averaged particle emission rate, $\bar{N}_{(1)}$, yielded a strong positive correlation (Fig. 1c), suggesting that a sizeable portion of the particulate matter might be dust aerosolized by animal movement rather than respiratory droplets. Consistent with particles originating from different sources, we identified a bimodal size distribution in measured particles (median diameters of 1.3 and 3.8 μm) for a guinea pig caged with its customary bedding of dried corncob granules (CC) (Supplementary Fig. 3a). Mobile guinea pigs emitted orders of magnitude more particles than stationary guinea pigs, regardless of whether their standard CC bedding was replaced with a polar fleece-covered absorbent pad (PF) or removed completely (Fig. 1e).

We next sought to measure airborne particulates exhaled directly from the respiratory tracts of three guinea pigs. To contain non-respiratory particulates like dander, fur, and dust emanating from the bodies of the animals themselves, we anesthetized each guinea pig and placed it in a closed aluminum sleeve, with only a small aperture for its nose (Fig. 1d; Supplementary Fig. 4). The resulting particle emission dynamics for the stationary animals were qualitatively different from the mobile animals; no spikes were observed, and the emission rates overall were considerably smaller (Supplementary Fig. 2b). Examination of the 15-min time-averaged particle emission rate, $\bar{N}_{(15)}$, revealed that anesthetized, stationary guinea pigs within the aluminum sleeve emitted 0.10–0.18 particles s$^{-1}$ prior to inoculation with influenza virus (day 0 post-inoculation, Fig. 1f), a 10-to-100-fold reduction in the average number of particles s$^{-1}$ produced by the same guinea pigs while awake and moving around in the cage on CC, PF, or no bedding. Intranasal inoculation with influenza A/Panama/2007/1999 (H3N2) (Pan99) virus only slightly increased particle emission by anesthetized, stationary guinea pigs, with up to 0.5 particles s$^{-1}$ measured on days 2–3 post-inoculation from two of the three animals (Fig. 1f). In contrast to mobile guinea pigs, the size distribution of the particles emitted by stationary animals was weighted toward the smallest size range. Approximately 11% of the particles emitted from a cage containing an awake, mobile guinea pig were 0.3–0.5 μm in diameter (Supplementary Fig. 3a), but in contrast ~58% of the particles emitted by the anesthetized animals were in the same size range (Supplementary Fig. 3b), similar to the proportion of 0.3–0.5 μm particles in the exhaled breath of humans[23].

Although this finding suggests that guinea pigs emit expiratory particles with a size range comparable to that of humans, we performed a negative control to validate this interpretation. All three animals were humanely euthanized, and APS measurements were repeated with the deceased animals in the aluminum sleeve. Unexpectedly, we found that, even in the absence of respiration, euthanized guinea pigs still emitted 0.07–0.2 particles s$^{-1}$ (Fig. 1f), similar to the rate observed for anesthetized, uninfected animals, but greater than the background rate in the absence of an animal (<0.005 particle s$^{-1}$). Likewise, the size distributions of the particles emitted by euthanized animals were similar to the anesthetized animals, with 69% of the particles between 0.3 and 0.5 μm (Supplementary Fig. 3c). Thus, neither the absolute number nor size distributions of the particles emitted by a guinea pig were appreciably different, regardless of whether the animal was alive and tidally breathing or had been euthanized. By enclosing the animals in an aluminum sleeve, with only their noses exposed, we attempted to contain all non-expiratory aerosols. However, despite these efforts to eliminate dust and dander from our measurements, a non-negligible fraction of the particles that initially appeared to be respiratory emissions actually were not directly exhaled particles.

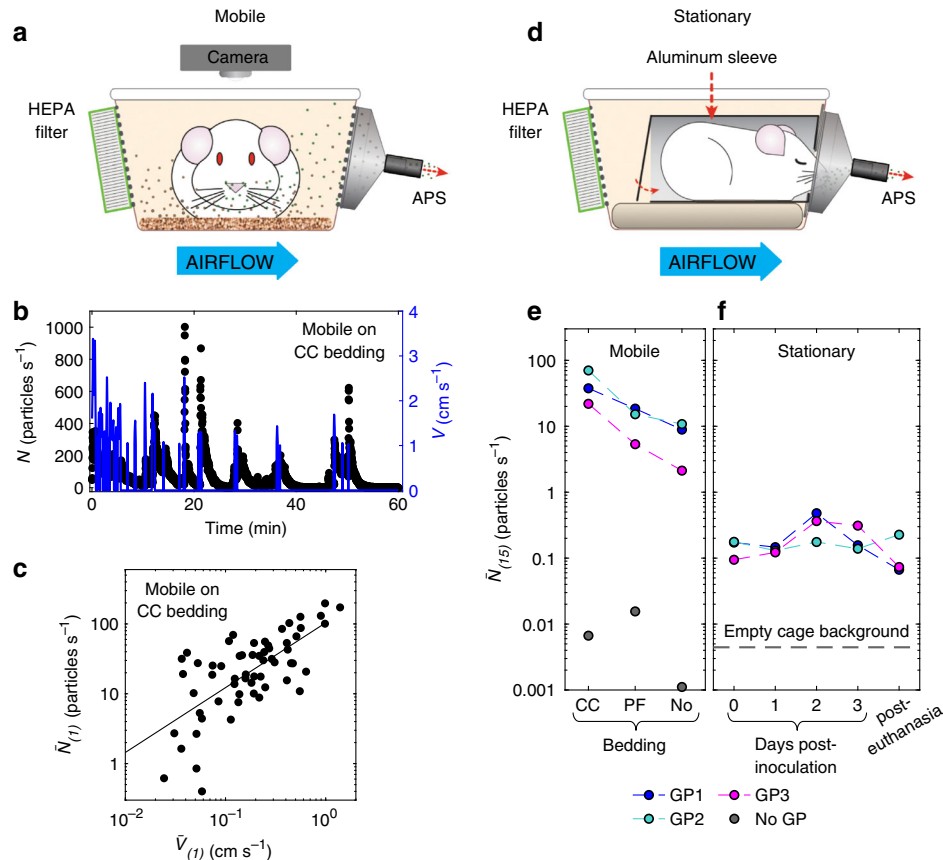

**Fig. 1 Guinea pigs aerosolize more dust than respiratory particles. a** Schematic for aerodynamic particle sizer (APS) experiments to quantify the airborne particulates generated by awake, unrestrained (mobile) guinea pigs (GP) (Supplementary Fig. 1). **b** Representative instantaneous particle emission rate (left axis) and instantaneous guinea pig movement velocity (right axis) vs. time for a mobile guinea pig in a cage with granular dried corncob (CC) bedding. **c** Time-averaged particle emission rate over 1 min ($\bar{N}_{(1)}$) vs. time-averaged guinea pig movement velocity over 1 min ($\bar{V}_{(1)}$). Solid line is the power law fit with exponent 0.93, correlation coefficient 0.80, and *p*-value $9.6 \times 10^{-15}$. **d** Schematic for APS experiments to measure the particulates produced by anesthetized or euthanized (stationary) guinea pigs (Supplementary Fig. 4). **e** Particle emission rates, time-averaged over 15 min ($\bar{N}_{(15)}$), for three mobile guinea pigs (GP1, GP2, and GP3). Gray markers denote background particle counts without a guinea pig in the cage with different beddings (dried corncob granulas (CC), polar fleece-covered absorbent pads (PF), or no bedding (No) on the plastic cage floor). **f** Measurements of the particle emission rates, time-averaged over 15 min ($\bar{N}_{(15)}$), for stationary guinea pigs, performed prior to inoculation (day 0) and on days 1, 2, and 3 post-inoculation with influenza A/Panama/2007/1999 (H3N2) (Pan99) virus, and after euthanasia. Horizontal gray dashed line denotes background particle counts of empty cage. Particle emission rates are the total of all particles detected in the size range of 0.3–20 μm in diameter (Supplementary Figs. 2 and 3). Source data are provided as a Source Data file.

**Transmission of influenza A virus via aerosolized fomites.** Given that environmental dust comprised such a large fraction of the total airborne particulates emitted by experimental guinea pigs, we hypothesized that, if these airborne environmental dust particulates were to become contaminated with influenza virus, they could serve as vehicles on which influenza virus might transmit through the air. We call these virus-contaminated dust particles "aerosolized fomites," to differentiate them not only from virus-laden respiratory droplets that are exhaled, coughed, or sneezed into the air by an infectious person or animal, but also from the macroscopic virus-contaminated objects that are traditionally thought of as fomites.

To investigate this hypothesis, we first infected guinea pigs with Pan99 influenza virus by intranasal inoculation using a standard protocol, and then we assessed the degree of environmental contamination in their cages over time. We found that swab samples from their fur, ears, paws, and cages all yielded viable virus by 2 days post-inoculation (dpi), and virus remained cultivable from swabs until at least 3 dpi (Fig. 2a). No measurable

virus was cultured from swabs taken either on the day of inoculation or on 1 dpi, demonstrating that virus replicating within the respiratory tract, rather than the initial inoculum, was being spread to and persisting on their bodies and environment. We noted typical behaviors demonstrated by guinea pigs, such as grooming and nose rubbing (Fig. 2b; Supplementary Movie 1), that may have contributed to the spread of the virus to the animal's body and environment.

We next determined whether airborne influenza virus transmission could occur from a virus-contaminated environment, in the absence of viral replication in the donor animal's respiratory tract. To mimic the self-contamination that we had observed in intranasally inoculated animals (Fig. 2a), we applied Pan99 stock virus with a paintbrush to the bodies of guinea pigs that had been previously infected with Pan99 and thus were immune to re-infection[24]. We then paired the contaminated virus-donor animals with susceptible virus-recipient animals in cages that only permit transmission by airborne routes (Fig. 3a; Supplementary Fig. 5). No virus was detected in nasal washes

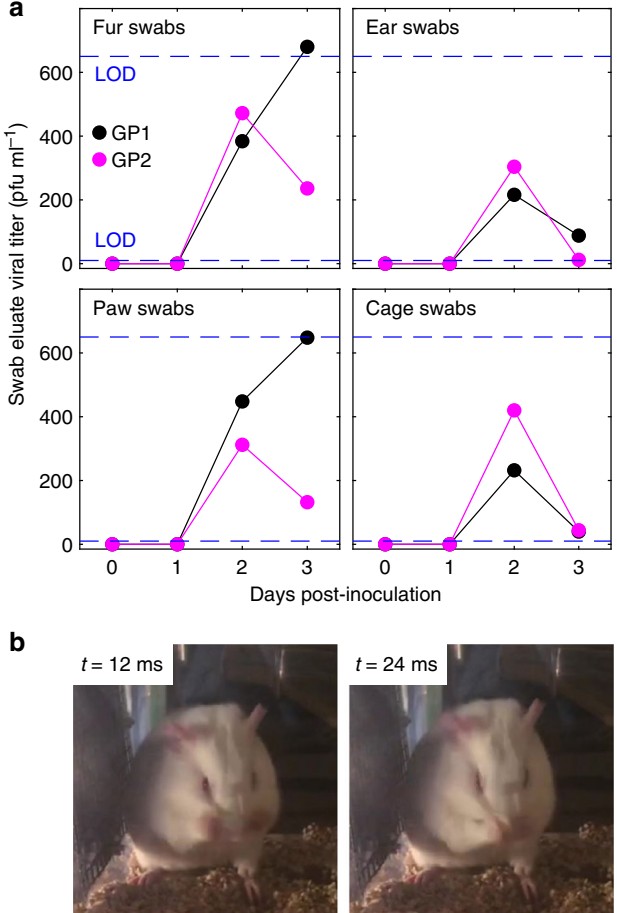

**Fig. 2 Infected guinea pigs contaminate their environments with viable influenza virus. a** Swab eluate viral titers (in plaque-forming units (pfu) ml$^{-1}$) from fur, ears, paws, and cages of two separately housed guinea pigs (GP1 and GP2) after intranasal inoculation with Pan99. Two biological replicates (two individual guinea pigs) were performed. One swab per area (fur, ears, paws, and cages) was taken, and one plaque assay per swab eluate was performed (one technical replicate per swab from each biological replicate). Horizontal dashed lines indicate upper (650 pfu ml$^{-1}$ of swab eluate) and lower (4 pfu ml$^{-1}$) limits of detection (LOD) for the titration plaque assays. **b** An intranasally inoculated guinea pig grooming (Supplementary Movie 1). Source data are provided as a Source Data file.

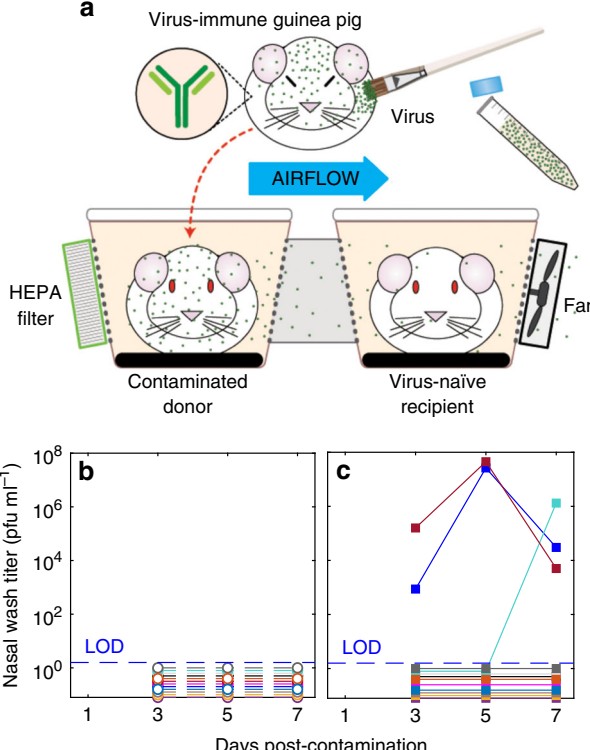

**Fig. 3 Influenza virus-naive guinea pigs are infected by aerosolized fomites. a** Transmission experiment schematic, showing a virus-naive recipient guinea pig placed downwind of, but physically separated from, a virus-immune, virus-contaminated donor guinea pig. **b** Nasal wash virus titers, in plaque-forming units (pfu) ml$^{-1}$, from 12 immune, contaminated donor guinea pigs, each represented by a different color. **c** Nasal wash virus titers from 12 recipient guinea pigs. Each color represents an individual recipient. Dotted line indictes the limit of detection (LOD). Source data are provided as a Source Data file.

from any of the immune donor guinea pigs (Fig. 3b); however, we observed influenza virus transmission in 3 of 12 animal pairs (25% transmission rate, Fig. 3c). Swab samples from the bodies of the immune, virus-contaminated guinea pigs and their environment confirmed the presence of viable virus at days 2 and 4 post-contamination (Supplementary Fig. 6). Thus, we conclude that airborne particulate matter from a non-respiratory source is able to transmit influenza virus through the air to a susceptible host.

**Generation of aerosolized fomites from an inanimate source.** Finally, we explored the generation of infectious aerosolized fomites from a virus-contaminated but inanimate dust source. We applied stock Pan99 virus in liquid solution to various commercially available paper tissues and towels and let them dry thoroughly in a biosafety cabinet. Crumpling, folding, and rubbing the dried paper tissues by hand released up to ~900 particles s$^{-1}$ as measured by the APS (Fig. 4a, b; Supplementary Movie 2). The size distribution of the tissue-generated airborne

particulates was in the respirable range (Supplementary Fig. 3d), with 99.8% of the particles in the range 0.3 to 10 µm, similar to those generated by guinea pigs moving in their cages (Supplementary Fig. 3a). After 8 min of crumpling paper tissues by hand and collecting the aerosols with a bioaerosol sampler[25], plaque assay titration of the collection media from the air sampler demonstrated that these aerosolized fomites contained cultivable influenza virus, which was captured at a rate of 1–5 pfu min$^{-1}$ of air sampling (Fig. 4c; Supplementary Fig. 7). We conservatively estimate that only 0.03% of the mass of the contaminated paper tissue was actually aerosolized into the bioaerosol collector, yielding a maximum estimated virus release rate of 14 pfu min$^{-1}$ of tissue manipulation, similar to that observed experimentally (see Supplementary Discussion 1 for details). Because the bioaerosol sampler is designed to preserve virus infectivity rather than discriminate by size, any particles within the range of 0.3 to 10 µm may have carried viable virus, and we cannot draw any further conclusions about particle size and viral payload. Nevertheless, our experiments do demonstrate that an influenza virus-contaminated tissue, dried under typical indoor environmental conditions, retained its infectiousness and, upon handling, released viable influenza virus into the air, carried on airborne particulates in the respirable range.

## Discussion
These results show that dried influenza virus remains viable in the environment, on materials like paper tissues and on the bodies of

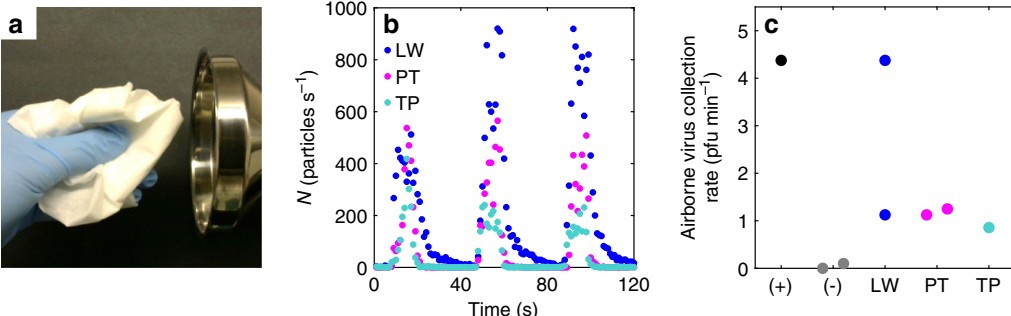

**Fig. 4 Paper tissues contaminated with Pan99 virus release infectious aerosolized fomites. a** Manually rubbing paper tissues in front of a stainless-steel funnel attached to the aerodynamic particle sizer (APS) (see also Supplementary Movie 2). **b** Instantaneous particle emission rate vs. time, measured by APS, produced by rubbing three different types of paper tissues, lab wipe (LW), paper towel (PT), and toilet paper (TP). The positive control (Pan99 virus at 200 pfu ml$^{-1}$, aerosolized by a nebulizer directed into the funnel) is indicated by (+), and the negative control (air sampling without virus aerosolization) is indicated by (-). Rates represent the total of all particles detected in the size range of 0.3–20 μm (Supplementary Fig. 3). **c** Quantification by plaque assay of viable airborne viruses (pfu min$^{-1}$ of air sampling), collected while rubbing virus-contaminated paper tissues in front of a BioSpot bioaerosol sampler (Supplementary Fig. 7). Source data are provided as a Source Data file.

living animals, long enough to be aerosolized on non-respiratory dust particles that can transmit infection through the air to new mammalian hosts.

In vitro, we demonstrate that an influenza A virus, dried on paper tissues for 30–45 min at ambient room temperature and humidity, retained infectiousness on aerosolized fomites that were collected from the air with a bioaerosol sampler and subsequently grown in cell culture. The quantity of aerosolized fomites generated by this method, 1–5 pfu min$^{-1}$ of air sampling, appears relatively small, both in absolute number and relative to the total amount of airborne particles produced. However, a recent study of the exhaled breath of symptomatic human volunteers with influenza[20] found that 39% of participants exhaled culturable virus in particles sized 0.05 to 5 μm, at a geometric mean virus titer of 37 fluorescent focus units (FFU) per 30-min sample (1.2 FFU min$^{-1}$), similar to our tissue experiments; the highest exhalation rate was 1100 FFU per 30 min (37 FFU min$^{-1}$)[20]. Ultimately, the clinical import of any quantity of aerosolized fomites depends entirely on their infectiousness in the susceptible human host who inhales them. The human infectious dose for Pan99 aerosolized from paper tissues is unknown, but for an influenza A (H2N2) virus in liquid solution, aerosolized into similarly sized particles (1–3 μm), inhalation of ~2 pfu or fewer (0.6 to 3 TCID$_{50}$) was found to be sufficient to initiate human infection[26]. Thus, we conclude that human infection by aerosolized fomites generated from an inanimate source like a virus-contaminated tissue is possible, though it remains to be demonstrated experimentally or empirically.

In vivo, we show that guinea pigs painted with influenza virus harbored viable virus on their bodies for up to 4 days post-contamination, which was subsequently transmitted through the air to infect 3 of 12 virus-susceptible partner animals housed in separate cages (25% transmission rate, 95% credible interval 8–52%). This transmission rate is lower than previously observed, under similar environmental conditions, with the same influenza virus isolate after intranasal inoculation into donor animals: 7 transmission events in 8 pairs of guinea pigs[27] (88% transmission rate, 95% credible interval 56–99%) and 2 transmission events in 3 pairs of ferrets[28] (67% transmission rate, 95% credible interval 23–96%). These results suggest that influenza virus transmission via aerosolized fomites may be less efficient than transmission by respiratory droplets or droplet nuclei, under the conditions tested. An alternate interpretation, however, is that the immune, virus-painted donor guinea pigs contaminated their environment less effectively than the intranasally inoculated guinea pigs, thus

decreasing the number of aerosolized fomites that could be produced under these conditions. The intranasally inoculated animals had the opportunity to continuously re-contaminate their environment as the virus replicated in their respiratory tracts, before their immune systems suppressed it. In contrast, the immune, virus-contaminated animals were contaminated only once, and viable influenza virus was not replenished. While the swabs from the animals' bodies yielded comparable viable virus titers on days 2 and 4 post-inoculation or post-contamination, in the range of hundreds of pfu ml$^{-1}$ of swab eluate, we recovered substantially more viable influenza virus from the cage walls of the intranasally inoculated animals than the immune, contaminated ones, a discrepancy that is as yet unexplained. Virus deposition by inoculated guinea pigs, possibly via direct nose and mouth contact, may be more efficient than the transfer of dried virus from the bodies of contaminated guinea pigs or its retention on the cage walls. Respiratory mucus may also have a preservative effect on non-porous surfaces, such that deposited virus retains its infectivity longer. These hypotheses remain open for future research. However, our results do serve as a clear proof of principle that influenza virus can transmit through the air when carried on micron-sized, non-respiratory particulate matter from the environment.

Influenza virus transport on dust has been infrequently hypothesized[4,17,29] and experimentally explored[30] in the past, and other respiratory pathogens are known or suspected to initiate human infection in this manner[31,32]. However, current opinion about influenza virus transmission, both in humans[7–11] and in animal models[16,33], appears to presume that airborne infectious virus derives solely from exhaled, coughed, or sneezed respiratory particles, or occasionally from aerosol-generating medical procedures[6,10,11]. We now provide direct experimental evidence that the airborne particles mediating mammalian influenza virus transmission need not be directly emitted into the air from the respiratory tract of an infectious host; rather, aerosolized fomites from a virus-contaminated environment can also spread influenza viruses through the air.

As we have demonstrated in guinea pigs, infected ferrets have been shown to contaminate dust in their environment with infectious influenza virus[30], raising the possibility that aerosolized fomites, and not solely expiratory particles, may contribute to airborne virus transmission in other animal species. Though only experimentally demonstrated in the guinea pig model of influenza virus transmission to date, aerosolized fomites could plausibly contribute to airborne transmission in other animal models,

including not only ferrets but also hamsters[34,35], swine[36,37], and mice[38,39]. This possibility should be experimentally explored in a rigorous fashion, because the confidence with which we can extrapolate animal data to human influenza rests on the degree to which animal models are representative of human influenza. If aerosolized fomites contribute differently to airborne influenza virus transmission in animal models and humans, it would be an important caveat in generalizing animal data to public health.

Our data do not explain why certain strains of influenza virus transmit with poor efficiency via airborne routes, but the literature offers some clues. Avian influenza A viruses of the H5N1 subtype have been found to replicate efficiently in but transmit poorly among both ferrets and guinea pigs, and key amino acids in a few viral proteins have been implicated in determining airborne transmissibility in these animal models[40,41]. In particular, mutations that increase the thermal and acid stability of the receptor-binding hemagglutinin (HA) protein enhance the transmissibility of H5N1 viruses in ferrets[40,42]. Linster et al.[40] hypothesized that, although the mutations that enhanced "airborne transmission between ferrets may be related to the pH of fusion or thermostability, these properties may merely be a surrogate for another—as yet unknown—phenotype, such as stability of HA in aerosols, resistance to drought, stability in mucus, or altered pH in the host environment"[40]. Subsequently, when aerosolized under laboratory conditions, swine influenza viruses that transmit efficiently by air among ferrets were shown to remain infectious in aerosols longer than viruses that transmit less efficiently[43]. When the airborne virus source was an infected ferret, a human influenza virus isolate from the 2009 H1N1 pandemic demonstrated enhanced airborne survival, relative to a variant virus encoding a destabilizing HA mutation that increases its pH of fusion[44]. These data suggest that the stability of the HA protein in the environment is one critical factor in influenza virus transmissibility by airborne routes.

However, species also has a role in the transmissibility of different influenza virus strains. Guinea pigs do not transmit pre-2009 H1N1 viruses efficiently[45,46], while ferrets do[47]. Ferrets have been shown to transmit fewer strains of influenza B virus by airborne routes[48] than guinea pigs[49,50]. But both influenza A (H1N1) and B viruses have transmitted well enough among people to become endemic in humans. Given the overall complexity of influenza virus transmission, it is unlikely that the presence of aerosolized fomites or their environmental stability entirely explains why some influenza viruses transmit through the air efficiently while others do not. More likely, airborne transmission efficiency depends variably on multiple factors: the infectiousness of the virus donor, the susceptibility of the virus recipient, and the stability of the virus in the environment between them. The transmission chain has many links, any of which could be a weak link that precludes efficient transmission, whether by aerosolized fomites or any other route. Despite the many remaining uncertainties surrounding influenza virus transmission in both humans and animal models, however, our data do indicate that aerosolized fomites may contribute, under specific circumstances that remain to be fully elucidated, to airborne influenza virus transmission in an animal model.

To our knowledge, no experimental evidence exists to establish that the airborne transmission of influenza viruses between experimental animals, or even between humans, occurs entirely via exhaled respiratory particles, as is commonly presumed. Our experimental data confirm that influenza virus transmission by aerosolized fomites is, at minimum, biologically plausible, and possibly generalizable to other respiratory viruses that transmit preferentially or opportunistically[12] by the airborne route. During the COVID-19 pandemic in China, air sampling in various hospital locations found the highest airborne genome counts of SARS-CoV-2 in rooms where health care workers doffed their personal protective equipment (PPE), hinting that virus was possibly being aerosolized from contaminated clothing as it was removed[51]. In light of our experiments, we conclude that the contribution of aerosolized fomites to respiratory virus transmission in both humans and animal models requires further scientific consideration and rigorous investigation.

## Methods

**Viruses and cells.** Influenza A/Panama/2007/1999 (H3N2) virus (Pan99) was cultured in Madin Darby canine kidney (MDCK)-SIAT1 cells[52] (Millipore Sigma) in Dulbecco's Modified Eagle Medium (DMEM; Gibco) supplemented with 10% fetal calf serum and 1 mg ml$^{-1}$ of G418 selective antibiotic (Geneticin 100×, Gibco) at 37 °C and 5% $CO_2$[52]. Stock virus was grown in the allantoic fluid of 10-day-old embryonated chicken eggs for 48 h at 37 °C[53] and then frozen in aliquots at −80 °C.

**Animal experiments.** Five- to six-week-old female Hartley strain guinea pigs weighing 350–400 g were obtained from Charles River Laboratories. Animals were allowed access to food and water ad libitum and kept on a 12-h light-dark cycle. Prior to intranasal virus inoculation, guinea pigs were anesthetized with a mixture of ketamine (30 mg/kg of body weight) and xylazine (5 mg kg$^{-1}$) administered intramuscularly. For intranasal inoculation, Pan99 stock virus was diluted in PBS supplemented with antibiotics (Penicillin–Streptomycin 10,000 U ml$^{-1}$, Gibco) (PBS + P/S). An inoculum of $10^4$ plaque-forming units (pfu) in 300 µl was instilled intranasally by applying 150 µl to each nostril[27,54]. Inoculated guinea pigs were placed supine, in a nose-up position, during recovery from anesthesia (~30–45 min). All procedures were performed in strict accordance with the recommendations in the Guide for the Care and Use of Laboratory Animals[55], and the research protocol was approved by the Icahn School of Medicine at Mount Sinai Institutional Animal Care and Use Committee (IACUC protocol #2014-0178).

**Quantification of airborne particulates.** To measure airborne particulates generated by an awake, unrestrained guinea pig, a standard polycarbonate animal cage (26.7 cm × 48.3 cm × 20.3 cm) with an airtight, transparent Plexiglas lid, was modified to be connected to an aerodynamic particle sizer (APS, Model 3321, TSI Inc.) with conductive silicone tubing (inner diameter of 0.95 cm and length of 40 cm) (Fig. 1a). The APS detects, counts, and measures with high accuracy the sizes of particles with aerodynamic diameters between 0.5 and 20 µm; for smaller sizes, between 0.3 and 0.5 µm, the APS detects and counts the particles but cannot distinguish their size distribution in this range. APS data were recorded with Aerosol Instrument Manager (AIM) software, version 9.0.0.0 (TSI Inc.). Air was drawn into the cage through two high efficiency particulate air (HEPA) filters (Dirt Devil F66, TTI Floor Care North America), each 10.2 cm × 10.2 cm × 4.3 cm, by the APS pulling air at 5 L min$^{-1}$. The guinea pig's customary food pellets and drinking water were provided in the cage during measurements. An ultra-wide-angle web camera (Genius WideCam F100, KYE International Corp.) mounted above the cage recorded guinea pig movement at 1 image s$^{-1}$ using iSpy (64-bit) software, version 7.2.1.0. Three healthy guinea pigs were placed individually inside the measurement cage with granulated dry corncob (CC) bedding (Supplementary Fig. 1a); with custom-made polar fleece- (PF-) covered[56] disposable absorbent pads (Fisherbrand Universal All-Purpose Absorbent Pads, Fisher Healthcare) (Supplementary Fig. 1b); or without bedding (Supplementary Fig. 1c). The particles emitted from the guinea pig cage were measured by APS at 1-s intervals over 1 h for each guinea pig. Custom code, written in MATLAB (MathWorks), identified the guinea pig centroid in each time-lapse image and calculated the guinea pig's velocity by quantifying displacement in the centroid's location over each 1-s interval. To test for correlation between particle production and animal motion, we time-averaged the particle emission rate ($\bar{N}_{(1)}$) and guinea pig movement velocity ($\bar{V}_{(1)}$) over 1-min periods, as denoted by the subscript "(1)" (Fig. 1c). To assess the particle production by mobile guinea pigs on different beddings (Fig. 1e), the particle emission rates were measured over a total sampling period of 1 h and then time-averaged over four 15-min periods, $\bar{N}_{(15)}$. The particle emission rate reported for mobile guinea pigs in Fig. 1e is the average of four time-averaged particle emission rates ($\bar{N}_{(15)}$). A wash-out to remove background particulates was performed prior to placing the guinea pigs in the measurement cage but was not practicable afterwards, as the guinea pigs were free to move around and generate airborne particulates as soon as they were placed in the cage. Background particle emission rate in the absence of the animals (gray circles, Fig. 1e) was measured in the cages without animals, after a wash-out.

The APS measurement cage was also used to quantify the expiratory particles from three anesthetized guinea pigs prior to and after intranasal inoculation with Pan99. To minimize non-respiratory background particulates, each anesthetized guinea pig was placed in a closed aluminum sleeve (Supplementary Fig. 4a), with its nose protruding from a small aperture (Supplementary Fig. 4b), which was then placed inside the airtight HEPA-filtered cage and attached with magnets directly to the stainless-steel funnel connected to the APS (Fig. 1d; Supplementary Fig. 4c, d). APS measurements were performed with each guinea pig individually, over 30 min,

both prior to virus inoculation (day 0) and on days 1, 2, and 3 post-inoculation. As a negative control, we humanely euthanized the same three guinea pigs by $CO_2$ inhalation under ketamine and xylazine anesthesia. Death was confirmed by physical examination[57] before placing them individually inside the aluminum sleeve. APS measurements were taken for 30 min per guinea pig. The time-averaged particle emission rates, $\bar{N}_{(15)}$, (Fig. 1f) were calculated from the final 15 min of the total 30-min measurement, using the first 15 min as a wash-out to remove background particulates introduced during placement of the anesthetized or euthanized animals in the measurement cage. The gray dashed line (Fig. 1f) indicating the background level was derived from the average of three 15-min measurements of the cage with no animal present.

Each APS experiment was performed once per condition, on each of three individual guinea pigs. For the awake, mobile guinea pigs, the variable experimental condition was bedding type (three different bedding types, one 1-h measurement per bedding type per guinea pig). For stationary guinea pigs, the variable in the experimental conditions were pre-infection vs. post-infection with Pan99 virus and anesthetized vs. euthanized guinea pigs. Measurements on anesthetized guinea pigs were performed on 4 different days (pre-inoculation and days 1, 2, and 3 post-inoculation, one 30-min measurement per day per guinea pig), and once with the euthanized guinea pigs (one 30-min measurement per guinea pig).

**Influenza virus transmission experiments.** Transmission experiments were performed at constant temperature and relative humidity (RH) in environmentally controlled chambers (model 6030, Caron Products & Services, Inc.) set at 20 °C and 20% RH. Each transmission experiment replicate comprised four pairs of guinea pigs, each with a virus-immune donor, contaminated with influenza virus, paired with a virus-naive recipient. Each transmission pair was housed in a custom-fabricated cage unit consisting of two standard polycarbonate animal cages (each 26.7 cm × 48.3 cm × 20.3 cm) joined together by a stainless-steel air conduit (7.6 cm × 34.3 cm × 14 cm). Two panels of wire mesh (1.25 cm² openings) closed off both sides of the air conduit to prevent contact between guinea pigs while allowing airflow. The air was drawn through the transmission cage unit in a unidirectional manner, from donor guinea pig upstream toward the recipient guinea pig downstream (Fig. 3a; Supplementary Fig. 5). In the cage housing the virus-donor animal, air entered the transmission cage unit through a HEPA filter (Aer1 HAPF300AH, Holmes Products) mounted over the air intake aperture (23 cm × 10 cm). In the cage housing the virus-recipient animal, two adjustable-speed fans (model FFC1212DE, Highfine Electronics, Inc.) exhausted air outward through the air outlet aperture (23 cm × 11.5 cm). Airflow was regulated by adjusting fan speed with a speed controller (Model ZS-X4B, Elegiant). A hot-wire anemometer probe (Alnor model AVM440, TSI Inc.) measured and recorded airflow velocity, temperature, and RH in the center of the air conduit. Airflow speed was set at 0.5 m s⁻¹ for all experiments. Polar fleece-covered absorbent pads (PF) were used for cage bedding, and guinea pig chow and water were also supplied in the cage.

Donor guinea pigs had been previously infected with Pan99 at least 6 weeks prior to these experiments. On day 0, 10 ml of Pan99 virus at a concentration of 10⁷ pfu ml⁻¹ was applied with a paintbrush to the bodies of the Pan99-immune virus-donor guinea pigs, which were then placed into the donor (upstream) compartment of a transmission cage unit and allowed to dry at regulated temperature (20 °C) and humidity (20% RH), with the cage fans off, prior to placing an influenza virus-naive guinea pig into the recipient (downstream) compartment. Recipient guinea pigs were kept in a separate room in the animal vivarium during the application of influenza virus to the immune donor guinea pigs, and gloves were changed before handling the recipient guinea pigs to place them into the transmission cage unit. Transmission pairs were kept together for a total of 7 days, and nasal washing was performed on days 2, 4, and 6 post-contamination by instilling a total of 1 ml of PBS + P/S into both nares and allowing it to drain onto a sterile Petri dish.

Samples were collected in 1.5-ml tubes on ice, centrifuged to pellet debris, and stored at −80 °C until titration by plaque assay. Between transmission experiments, polar fleece bedding covers were laundered through two complete hot-water washer cycles, first with a dye- and perfume-free detergent (Tide Free and Gentle Liquid Laundry Detergent, Procter & Gamble) and then again without detergent. Covers were tumble-dried at the high-heat setting and then placed over new absorbent pads (Fisherbrand Universal All-Purpose Absorbent Pads, Fisher Healthcare). Pads and polar fleece covers were autoclaved (15-min dry cycle) before use.

**Quantification of environmental virus contamination.** Assessment of the duration of virus viability on guinea pigs' bodies was performed by swabbing fur, ears and paws with sterile cotton tipped applicators (25-8061PC, Puritan Medical Products) wetted in 1 ml of PBS + P/S and then eluted in the same solution after swabbing. Cage walls were swabbed in the same manner. Swab elution samples were kept on ice until centrifugation to pellet debris and were then titrated by plaque assay immediately afterwards.

Intranasally inoculated guinea pigs were housed in cages with corncob (CC) bedding during the environmental swabbing experiment. Immune, virus-contaminated guinea pigs were swabbed during the transmission experiments in

which they were the virus-donor animals. They were housed in the transmission cage units with polar fleece (PF) bedding, as described above.

**Aerosolized fomite generation and quantification.** The APS was used to measure the emission rate and size distribution of aerosolized fomites between 0.3 and 20 μm in diameter generated from three different kinds of paper tissues, including lab wipes (Kimwipes Delicate Task Wipers, Kimtech Science, Kimberly-Clark), paper towels (Scott, Kimberly-Clark), and toilet paper (Envision, Georgia Pacific). A stainless-steel funnel was connected to the APS inside a biosafety cabinet (BSC) with conductive silicone tubing (inner diameter of 0.95 cm and length of 15 cm) and 2-min samples were collected; each tissue was rubbed for 10 s in front of the funnel followed by 30 s rest, repeated three times, for a total of 30 s of particle generation (Supplementary Movie 2).

**Infectious aerosolized fomite collection and quantification.** To generate infectious aerosolized fomites, Pan99 stock virus was diluted in PBS + 0.5% BSA + P/S/A and was added drop-wise to lab wipes, paper towels, and toilet paper, at $3.6 \times 10^5$ pfu per whole tissue. After contamination, tissues were completely dried in the BSC. After drying, aerosolized fomites were generated by crumpling, rubbing, and folding the tissue by hand. The tissue was periodically readjusted to spread the manipulation over the entire surface area of the tissue. A prototype bioaerosol collector[25,58,59] (BioSpot, Aerosol Devices, Inc., Fort Collins, CO) was used to collect and enumerate infectious aerosolized fomites. The BioSpot collects particles in the size range of 10 nm to 10 μm, without differentiation by size, by enlarging particles through condensation of water onto them. We connected a stainless-steel funnel to the BioSpot bioaerosol collector with conductive silicone tubing (inner diameter of 0.95 cm and length of 80 cm). The funnel was placed in the BSC for each 10-min sampling period. Tissues were rubbed in front of the funnel for 8 min, followed by a 2-min wash-out with air drawn from the BSC in the absence of particle production. The positive control was Pan99 virus in PBS + 0.3% BSA + P/S/A (200 pfu ml⁻¹) aerosolized at 0.42 ml min⁻¹ by a nebulizer (Aeroneb Lab Small VMD Nebulizer, Aerogen) positioned in front of the funnel, and the negative control was air drawn from the empty BSC. The BioSpot air sample flow rate was 8 L min⁻¹, and airborne particles, enlarged via condensation, were collected into 2 ml of phosphate-buffered saline (PBS) supplemented with 0.5% bovine serum albumin (BSA; MP Biomedicals) and penicillin, streptomycin, and amphotericin (P/S/A; Antibiotic-Antimycotic 100×, Gibco) in 35 mm polystyrene Petri dishes (VWR). After collection, samples were transferred into sterile 5-ml tubes and kept on ice until titration.

**Quantification of viable influenza virus.** The virus titers of Pan99 stock aliquots and guinea pig nasal washes and body swab eluates were determined by plaque assay of 10-fold serial dilutions on MDCK-SIAT1 cells, grown in confluent monolayers in 6-well plates[60]. Viruses were serially diluted in PBS + 0.3% BSA + P/S/A, inoculated onto cell monolayers, and then incubated at 37 °C and 5% $CO_2$ for 1 h. After incubation, monolayers were overlaid with 1.3% microcrystalline cellulose (Avicel RC-591, FMC Biopolymer) in plaque assay medium, consisting of Minimum Essential Medium (MEM) supplemented with 1× Ham's F-12K nutrient mix, 1.2% sodium bicarbonate, 100 mM HEPES buffer, 20 mM L-glutamine, and P/S/A (all from Gibco), with 0.3% BSA and 1 μg ml⁻¹ of tosylsulfonyl phenylalanyl chloromethyl ketone (TPCK)-treated trypsin (Thermo Scientific). Plates were incubated at 37 °C and 5% $CO_2$ for 3 days. Cells were then fixed with 4% formaldehyde, and plaques were visualized with a crystal violet counterstain. BioSpot bioaerosol collection samples were titrated, undiluted, by inoculation onto MDCK-SIAT1 cells that had been grown to confluence in 35-mm-tissue culture dishes (two dishes per sample, 800 μl per dish). Dishes were spinoculated (300 × g) for 60 min at room temperature, in stacks of four dishes, in the A-4-44 rotor of an Eppendorf 5804R centrifuge. The inocula were then aspirated, and dishes were overlaid with plaque assay medium, incubated, fixed, and stained as described above. Viral titers calculated from plaque assay counts were compiled in Excel for Mac 2011 version 14.7.3 (Microsoft Corporation) for import into MATLAB (version R2019a, MathWorks).

**Statistics.** The power law fit of the APS-quantified, time-averaged particle emission rate over 1 min ($\bar{N}_{(1)}$) and the time-averaged guinea pig movement velocity over 1 min ($\bar{V}_{(1)}$), and the calculation of the correlation coefficient and Pearson's $p$-value (Fig. 1c), were performed in MATLAB. Bayesian methods were employed to estimate the posterior 95% probability intervals for Pan99 transmission by different routes under similar environmental conditions in animal models, given the past data obtained from intranasally inoculated donor guinea pigs[27] and ferrets[28], and new data obtained from immune, virus-contaminated guinea pigs (Fig. 3b, c). Data were analyzed in R (version 3.6.3, R Foundation for Statistical Computing) with the R packages rjags[61,62], runjags[63], and HDInterval[64]. An agnostic beta prior (shape parameters $A = 1$ and $B = 1$), and a Bernoulli likelihood function were used to obtain a 95% credible interval for the posterior distribution of the transmission probability $\theta$[64] given each of these data sets.

**Reporting summary**. Further information on research design is available in the Nature Research Reporting Summary linked to this article.

## Data availability

All data are available in the manuscript or the Source Data file. Source data are provided with this paper.

## Code availability

MATLAB code for quantifying guinea pig movement velocity from time-lapse images is available from the corresponding author upon reasonable request.

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

## Acknowledgements

We thank Pat Keady and Aerosol Devices Inc. for building the prototype BioSpot bioaerosol collector and for subsequent technical support. We also thank Mathew Komen for technical assistance. This research was supported by the National Institute for Allergy and Infectious Diseases (NIAID) of the National Institutes of Health (NIH), grant R01 AI110703 to N.M.B., W.D.R., and A.S.W.

## Author contributions

S.A. performed the APS measurements of airborne particulates emitted from the guinea pig cages and released by manipulation of paper tissues by hand. N.G., R.S.B., and S.A. performed guinea pig experiments, including intranasal virus inoculations, transmission experiments with virus-contaminated donor guinea pigs, and collection and titration of nasal washes and environmental swab samples. N.M.B. performed the BioSpot quantification of viable airborne viruses released from tissues. A.S.W., W.D.R., and N.M.B. conceived the project, and all authors contributed to experimental design. S.A., W.D.R., and N.M.B. analyzed the data and wrote the manuscript, assisted by N.G. with the Methods. All authors reviewed and revised the manuscript for accuracy and intellectual content.

## Competing interests

The authors declare no competing interests.
