## [Peer Review File · Nature Communications]

REVIEWER COMMENTS, first round:

Reviewer #1 (Remarks to the Author):

In this manuscript, Sima Asadi and colleagues investigated the presence of infectious influenza A virus in aerosolized fomites and the potential of such fomites to result in virus transmission between guinea pigs via the air. This is an important and interesting area of research, but the manuscript has a few problems:

1. The quantitative aspects of the work are not detailed properly.
2. Some observations in animal studies are not discussed well.

With respect to the first point:

A. For all particle measurements, an APS is used with detection range of 500-20000 nm. As a consequence, the lower range of particle sizes that may contain influenza virus (100-500 nm) is missed. This is particularly problematic given that (at least in humans) the vast majority of exhaled aerosols appear to be <500 nm, see eg Fabian et al., Plos ONE 2008. Thus, the quantitation of exhaled particles (e.g. in figure 1E) is potentially incorrect for a major fraction of the aerosols of importance (<500nm). The number of counted particles is mentioned in the text (around line 75) but not the corresponding particle sizes and the fact that small particles are missed. The number of exhaled small particles is not known for guinea pigs, as far as I am aware. With the use of the Biospot collector, ALL infectious particles are quantified (large plus small), making it very difficult, if not impossible, to allow quantitative conclusions.

B. The experimental procedures need to be much better identified. For instance, the authors show that the cage bedding determines – in part – the generation of aerosolized fomites, but the cage bedding is not specified (e.g. in figure 1) and as a consequence, the experiment can not be interpreted.

C. The scale bars in the different figures are not the same, making interpretation difficult, if not impossible. Fig 1B shows that up 1000 particles are detected per second (60000/min) but then Fig 1C shows a scalebar no larger than 10000/min (~166/sec) for the same(?) data and Fig 1E shows a scalebar with 0.1-0.5 particles per second (6-30/min). In Fig S1, which supposedly matches Fig 1B/1C, I see 0-70 particles/sec, orders of magnitude different between the two figures. The unit of measure should be the same throughout and an explanation should be provided for the difference between Fig S1 and 1.

D. In Fig 4, 500-1000 particles per second are generated (30000-60000/min), which yield 1-5/min infectious units, again with different units of measure.

E. In Fig S5, more infectious virus appears to have been loaded on the guinea pig fur than detected upon grooming, pushing the likelihood of virus transmission.

F. Although infectious aerosolized fomites can be created with tissues and towels, the fraction of infectious particles (1-4 per minute) is miniscule compared to 30,000-60,000 particles per minute that are created with relatively fresh (30-45 min) large amount (3.6×10^5) of virus.

On the whole, these quantitative aspects of the study require attention and a more thorough discussion.

With respect to the second point, the authors did not discuss the fact that many influenza viruses are NOT transmitted in animal models (e.g. ferrets) although the viruses clearly replicate in these hosts. H5N1 viruses and other zoonotic viruses replicate extensively in the ferret model and would thus contaminate the ferret (by grooming) and the cage without evidence of transmission via aerosol or respiratory droplets, whereas seasonal and pandemic influenza viruses are readily transmitted via this route. This validation of the model (e.g. summarized in Linster et al., Cell 2014), copying what is generally observed in real life, is not discussed in the manuscript. The authors challenge the validity of the ferret model (L164-167) but without providing an explanation for the lack of transmission of replication-competent viruses in this model. Some research groups even go as far as demonstrating contact transmission between ferrets or guinea pigs (through contaminated fomites, grooming, etc) without evidence of transmission via aerosol or respiratory droplets. This appears to contradict the importance of transmission via aerosolized fomites.

Additional comments:

In Fig 2A, LOD lines are misplaced (at least in my copy).

In L227-233, the authors highlight the background particle counts that are well known in the field and describe a “wash out” that is commonly used to deal with this problem in aerosol science. Was this wash out only done here, and not in other experiments?

Reviewer #2 (Remarks to the Author):

A. Summary of the key results

1. Virus-contaminated resuspended particles can transmit influenza in the guinea pig model.

B. Originality and significance

This work is highly original and significant because it shows that a previously underappreciated route of transmission—deposition of virus onto surfaces followed by resuspension and subsequent exposure to these—is possible.

C. Data, methodology, statistics

The methods are valid.

D. Conclusions

The conclusions are justified. The researchers showed, through multiple lines of evidence, that virus-containing particles are resuspended by physical motion and that exposure to them can initiate infection.

E. Clarity and context

The writing, figures, and overall presentation are very clear. The authors cite prior work appropriately to place this study in context.

1. line 45: “Uncertainty surrounding the modes by which influenza virus transmits among humans under different conditions hinders the development of interventions, like vaccines, designed to prevent influenza’s spread.” I can see how uncertainty hinders the development of non-pharmaceutical interventions, but the link to vaccine development is not obvious to this reader.

2. Figure 4: Images of the plaque assay plates are not needed.

Linsey Marr

Response to Reviewers of “Influenza A is Transmissible via Aerosolized Fomites,” by Asadi et al. (NCOMMS-20-08163)

Detailed responses are listed in order below.

Reviewer #1

In this manuscript, Sima Asadi and colleagues investigated the presence of infectious influenza A virus in aerosolized fomites and the potential of such fomites to result in virus transmission between guinea pigs via the air. This is an important and interesting area of research, but the manuscript has a few problems:

- 1. The quantitative aspects of the work are not detailed properly.*
- 2. Some observations in animal studies are not discussed well.*

We thank the reviewer for assessing our area of research as important and interesting. We also appreciate the reviewer’s detailed, critical review, which highlighted several shortcomings that we had not appreciated in the original manuscript. We believe that these critiques have enabled us to improve its technical quality and clarity and have greatly strengthened the work overall.

With respect to the first point:

A. For all particle measurements, an APS is used with detection range of 500-20000 nm. As a consequence, the lower range of particle sizes that may contain influenza virus (100-500 nm) is missed. This is particularly problematic given that (at least in humans) the vast majority of exhaled aerosols appear to be <500 nm, see eg Fabian et al., Plos ONE 2008. Thus, the quantitation of exhaled particles (e.g. in figure 1E) is potentially incorrect for a major fraction of the aerosols of importance (<500nm). The number of counted particles is mentioned in the text (around line 75) but not the corresponding particle sizes and the fact that small particles are missed.

The reviewer makes an excellent point; we did not adequately describe the specifications of our aerodynamic particle sizer (APS) in the original manuscript. As we noted in the methods, the TSI model 3321 APS both counts and measures the aerodynamic diameter of particles in the 0.5-20 μm range. However, it also counts particles between 0.3 and 0.5 μm ; it just cannot size them, because they are smaller than the wavelength of its laser.

We originally failed to note in certain figures (e.g., Fig. 1 of the original manuscript) that particle emission rates were based on *all* detected particles, including those between 0.3 and 0.5 μm , which is also the smallest bin width reported by Fabian et al. To address the reviewer’s comment, we have added a new supplementary figure: Fig. S2a shows the instantaneous small- and large-particle emission rates for one unrestrained, mobile guinea

pig (the same data as Fig. 1b, separated into small (0.3-0.5 μm) and large (0.5-20 μm) particle size ranges). Fig S2b shows the instantaneous small- and large-particle emission rates for one anesthetized, stationary guinea pig, for comparison to the awake, mobile guinea pig.

When we analyzed the data by particle size, we made a new observation: Approximately 11% of the particles emitted from a cage containing an awake, mobile guinea pig were in the smallest size range (0.3-0.5 μm). In contrast, although the anesthetized animals emitted many fewer particles overall, their size distribution was more heavily weighted toward the smallest size range; approximately 58% of the particles emitted by anesthetized animals were 0.3 to 0.5 μm . Thus, movement-generated airborne particulate matter is, as one might expect, on average larger than exhaled particles. Interestingly, though, our negative control (a euthanized guinea pig) emitted about 69% of particles in the range of 0.3 to 0.5 μm . In other words, there was no appreciable difference in the absolute number or size distribution of particles emitted by a guinea pig, regardless of whether it was breathing or not.

To address these points, we have revised the size distributions in supplementary Fig. S3 to include a bin for particles in the range of 0.3 to 0.5 μm , as in Fabian et al. We also added two new distributions to Fig. S3, comparing particle size distributions emitted by anesthetized and euthanized guinea pigs. Finally, we have explicitly stated the size range of the particles under discussion throughout the manuscript text.

The number of exhaled small particles is not known for guinea pigs, as far as I am aware.

We agree, and we were extremely surprised to find that guinea pigs, when anesthetized, hardly emit any more particles than euthanized guinea pigs. This, we believe, is a significant new finding; in conjunction with our other data, it indicates that the vast majority of airborne particles carried between guinea pig cages are non-expiratory in origin. Even though we attempted to contain all non-expiratory aerosols by enclosing the animals in a conductive aluminum sleeve, leaving only their noses exposed, we still found that a non-breathing animal released comparable particles (both in number and size) as a breathing animal. Collectively our data suggest that the majority of airborne particles emanating from the cage of a mobile animal are likely to be environmental dust or dander stirred up by movement, rather than respiratory droplets or droplet nuclei.

We also show that animals may emit non-respiratory particulates, even when one takes what appear to be reasonable measures to contain them. We are unaware of any other published literature in which the putative “expiratory” emissions of a breathing animal were compared against a true negative control, a non-breathing animal. Our results indicate that such rigorous controls may be necessary to interpret experiments intended to measure “exhaled” particles from animals.

With the use of the Biospot collector, ALL infectious particles are quantified (large plus small), making it very difficult, if not impossible, to allow quantitative conclusions.

The reviewer raises a key point, which we agree we did not adequately address in the original manuscript. The BioSpot is designed specifically to preserve virus viability by enlarging airborne particles through water condensation. Size-separation of airborne particulates, while simultaneously preserving the viability of viruses carried in or on them, remains technologically challenging (e.g., Bekking et al., *Influenza Other Respi Viruses* 2019; 13: 564– 573).

Our results nonetheless do allow an extremely important qualitative conclusion: the number of viable viruses carried by aerosolized fomites is nonzero. Given that little attention is currently paid to the possibility of virus re-aerosolization, our first goal is to report that it happens at all.

Our data do provide bounds on the size range, however. We know from the APS measurements that 99.8% of the particles emitted by the paper tissues ranged between 0.3 and 10 μm , with more than 95% of the particles less than 2 μm (see supplementary figure S3). To address this point, we have added text that explicitly states the bounds on the size range of the aerosolized fomites that we generated and notes that the size of virus-carrying particles within these bounds remains an open question (lines 152-155).

We agree with the reviewer that an important future task is to further elucidate which particles within this size range carry the most virus, but such specific measurements, we believe, are beyond the scope of the current manuscript, which is to show, for the first time to our knowledge, that aerosolized fomites carry viable influenza virus capable of infecting a susceptible host.

B. The experimental procedures need to be much better identified. For instance, the authors show that the cage bedding determines – in part – the generation of aerosolized fomites, but the cage bedding is not specified (e.g. in figure 1) and as a consequence, the experiment can not be interpreted.

We thank the reviewer for pointing out this omission. In response to this point and others discussed below, we have added the specific cage bedding to the caption of figure 1, and we have expanded the methods so that the cage bedding used in each guinea pig experiment is now explicitly stated.

C. The scale bars in the different figures are not the same, making interpretation difficult, if not impossible. Fig 1B shows that up 1000 particles are detected per second (60000/min) but then Fig 1C shows a scalebar no larger than 10000/min (~166/sec) for the same(?) data and Fig 1E shows a scalebar with 0.1-0.5 particles per second (6-30/min). In Fig S1, which supposedly matches Fig 1B/1C,

I see 0-70 particles/sec, orders of magnitude different between the two figures. The unit of measure should be the same throughout and an explanation should be provided for the difference between Fig S1 and 1.

This is an excellent point. We agree that the original presentation of our data was obfuscated by changes in the units between the different figures. Although these are all particle emission rates, the different units stemmed in part from how we time-averaged the data. Fig. 1B shows the “instantaneous” particle emission rate measured at 1-second intervals by the APS, which greatly varies over time; there are long stretches when it is close to zero, interrupted occasionally by abrupt particle emission spikes concurrent with animal motion. Accordingly, the majority of data points on a plot of instantaneous particle emission rate versus instantaneous velocity are clustered near the origin, obscuring the key qualitative trend. To even out these second-to-second changes, we instead averaged the particle emission rate and the guinea pig’s velocity over 1-minute periods, thus yielding the graph in Fig. 1c. Likewise, for Fig. S1d in the original manuscript, we sought to show that the type of bedding influenced the rate of particles emitted by awake, mobile animals; however, given that particle emission rate varied greatly even from minute to minute, we felt that focusing on short time frames would give misleading results. We instead averaged the particle emission rate over an entire 60-minute period to more broadly characterize the average particle emission rates from mobile guinea pigs on different beddings.

We agree that we did not adequately emphasize this time-averaging in the original manuscript, which was confusing. As suggested by the reviewer, then, we have revised the manuscript to always use the same units of all emission rates (particles/second). Likewise, to clarify the differences in time-averaging, we introduced the nomenclature $\bar{N}_{(1)}$ and $\bar{N}_{(15)}$, where the bar indicates the time average and the subscript denotes the averaging time period in minutes. We believe this notation will eliminate ambiguity about the type of time averaging performed.

Furthermore, we appreciate the reviewer’s overarching point that it was difficult to compare the particle emission rates between figures representing different experimental situations. In addition to standardizing units of measurement and differentiating instantaneous vs. time-averaged rates, as discussed above, we have also revised Fig 1e to directly compare the particle generation by mobile (new Fig. 1e) and stationary (new Fig. 1f) guinea pigs. We chose $\bar{N}_{(15)}$ as the basis of comparison because guinea pigs can reliably be anesthetized, and thus remain perfectly still, for at least 30 minutes. Accordingly, we measured particle emissions from stationary (anesthetized) guinea pigs for a total of 30 minutes, the first 15 minutes of which served as a “wash-out” (discussed more below). The second 15 minutes were time-averaged and plotted in figure 1f. We believe this revised figure shows more clearly what the reviewer was looking for, and drives home a key point: the anesthetized animals emitted hardly more particles than the euthanized ones, at a rate one to two orders of magnitude smaller than when the animals were awake and mobile. To

clarify this point, we have also added corresponding new text to the Results (e.g., lines 70-72, 86-92) and Methods (e.g., lines 311-318, 332-338)).

D. In Fig 4, 500-1000 particles per second are generated (30000-60000/min), which yield 1-5/min infectious units, again with different units of measure.

In Fig. 4b we show the instantaneous particle emission rate (without time averaging). In Fig. 4c, however, we report a very different measurement, the number of plaque-forming units (pfu) recovered by plaque assay of the BioSpot collection media, per minute of sampling flow directed into the BioSpot. The BioSpot operates at 8 L/min, so we could have expressed these data per liters of air sampled, as is sometimes done (e.g., Bekking et al., cited above); however, we feel that pfu per minute of sampling enables easier comparison to important data from the literature (e.g., Yan et al., PNAS 2018;115(5):1081-1086, discussed in lines 167-171). To clarify this point, we have revised the caption of Fig. 4 to call attention to the different types of measurements and corresponding units. In response to reviewer #2, we also moved the images of the plaque assays to the supplementary material, providing more room in the main manuscript for additional discussion.

E. In Fig S5, more infectious virus appears to have been loaded on the guinea pig fur than detected upon grooming, pushing the likelihood of virus transmission.

We strongly agree with the reviewer that increasing the amount of viral contamination on the animals will “push” the likelihood of virus transmission. But we emphasize a more fundamental point: That the probability of airborne transmission can be pushed one way or another by altering the amount of virus on an animal’s fur, rather than in its respiratory tract, is a novel concept in and of itself. In the words of reviewer #2, the idea of influenza transmission via aerosolized fomites has been “underappreciated.”

Initially, the inoculated animals did not have any detectable virus on their bodies, unlike the immune guinea pigs that were purposely contaminated (Fig. 2A). By days two to four post-inoculation or post-contamination, however, both the intranasally inoculated animals and the immune contaminated guinea pigs had pfu counts on the same order of magnitude (hundreds of pfu/mL of swab eluate). A key difference between the two situations is that the intranasally inoculated guinea pigs are actively replicating influenza virus in their respiratory tracts, while the immune animals are not. The intranasally inoculated animals have the opportunity to continuously contaminate their environment with “fresh” virus for days before their immune systems suppress viral replication. The amount of viral contamination in the environment of the intranasally inoculated animals increases from zero, reaches a maximum, and then decreases thereafter. In contrast, the immune animals were contaminated only once, and there was no mechanism to reintroduce infectious virus later in the experiment. The amount of viral contamination starts at a maximum and can

only decrease with time. Comparing swab titers “head-to-head” at single points in time does not capture the very different dynamics of environmental contamination in these two experimental conditions. A more meaningful comparison would involve the time integral of the viral contamination over the entire experimental period; however, aside from our manuscript, there are few data in the literature to calculate such integrals. Throat or nasal titers are routinely assessed, but environmental viral contamination is not. We believe our results open up a new whole avenue of investigation.

What was very different between intranasally inoculated and immune contaminated animals was the degree of cage wall contamination. The intranasally inoculated animals deposited significantly more viable virus on cage walls, suggesting that either deposition on inanimate surfaces is either more efficient, or virus viability is longer preserved, when the guinea pigs are intranasally inoculated. In contrast, much less contamination of the cage walls was observed with the immune contaminated animals, possibly because the dried virus on their fur is inefficiently transferred to the wall, or because its infectiousness decays faster. These remain open questions for future research, but we believe that they are beyond the scope of this manuscript, which is to show that aerosolized fomites can transmit influenza virus.

To address the above points, we have added discussion to the manuscript (lines 189-203) addressing the differences in the viral loadings and temporal dynamics between the intranasally inoculated and immune contaminated animals.

F. Although infectious aerosolized fomites can be created with tissues and towels, the fraction of infectious particles (1-4 per minute) is miniscule compared to 30,000-60,000 particles per minute that are created with relatively fresh (30-45 min) large amount (3.6×10^5) of virus. On the whole, these quantitative aspects of the study require attention and a more thorough discussion.

This is also an important point, which we thank the reviewer for raising. We agree that, at first glance, 1 to 4 pfu/min seems miniscule compared to 60,000 particles per minute, and that 3.6×10^5 pfu seems like a relatively much larger amount of virus. We hope that the following quantitative discussion will put these measurements into context.

In these experiments, 3.6×10^5 pfu of influenza virus, in liquid solution, was applied to the paper tissues. Although we added a “large” amount of virus to the tissue, it was dispersed through the entire tissue. Crucially, only a very small fraction of the tissue mass was aerosolized by manipulation. Conservatively, from our data in Fig. 4b, we estimate that a maximum of 500,000 total particles are aerosolized per tissue experiment (i.e., a maximum of 900 particles/s over 8 minutes of manipulation). From Fig. S3d, we obtain the geometric mean diameter on a mass basis (i.e., average of the diameter cubed) as $7.3 \mu\text{m}$. Given that the average density of cellulose is 1.5 g/cm^3 , we estimate that a maximum of $1.02 \times 10^{-4} \text{ g}$ of paper tissue was aerosolized into the BioSpot during manual rubbing. A lab wipe weighs

0.47 g, so only 0.032% of the original tissue mass was aerosolized into the BioSpot. Assuming the virus was evenly distributed over the mass of the tissue, only 115 pfu of the original 3.6×10^5 pfu were actually aerosolized – the rest stayed behind with the tissue. Since we collected 8 to 40 pfu per tissue rubbing experiment, our estimate of 115 pfu released from the tissue suggests that the BioSpot efficiently captured a sizable fraction of the virus-carrying particles aerosolized from the tissue.

Whether or not this amount of airborne virus is miniscule depends on the subsequent infectiousness of the particles in humans, which is not currently known for this virus on tissues. However, with an influenza A (H2N2) virus in liquid solution, aerosolized into similarly sized particles (1-3 μm), inhalation of approximately 2 pfu or fewer (0.6 to 3 TCID50) was sufficient to initiate human infection (Alford et al., Proceedings of the Society for Experimental Biology and Medicine 1966; 122:800-804). Our experiments, both with the tissue rubbing and the animal motion, indicate that it takes very little time (on the order of a minute) to generate a quantity of particles capable of carrying that amount of pfu.

To address these points, we have added more quantitative discussion to the text (lines 149-152, 166-179), with a new section in the supplementary material (Discussion S1) with details of the above calculations.

With respect to the second point, the authors did not discuss the fact that many influenza viruses are NOT transmitted in animal models (e.g. ferrets) although the viruses clearly replicate in these hosts. H5N1 viruses and other zoonotic viruses replicate extensively in the ferret model and would thus contaminate the ferret (by grooming) and the cage without evidence of transmission via aerosol or respiratory droplets, whereas seasonal and pandemic influenza viruses are readily transmitted via this route. This validation of the model (e.g. summarized in Linster et al., Cell 2014), copying what is generally observed in real life, is not discussed in the manuscript. The authors challenge the validity of the ferret model (L164-167) but without providing an explanation for the lack of transmission of replication-competent viruses in this model. Some research groups even go as far as demonstrating contact transmission between ferrets or guinea pigs (through contaminated fomites, grooming, etc) without evidence of transmission via aerosol or respiratory droplets. This appears to contradict the importance of transmission via aerosolized fomites.

The reviewer brings up an excellent point, and we agree our experiments answer some questions while raising many more. The reviewer states that the ferrets contaminate themselves by grooming. Although this hypothesis is very plausible, it does not appear that anyone since the 1940s has actually measured environmental contamination by influenza virus-infected ferrets during airborne transmission experiments. We know of no data demonstrating that ferrets infected with influenza A(H5N1) or other modern strains do or do not contaminate their fur and environment with influenza virus. We know relatively much more about environmental contamination by influenza virus-infected humans (e.g.,

Simmerman et al., *Clin Infect Dis* 2010; 51(9):1053-61; Mukherjee et al., *Am J Infect Control* 2012;40(7):590-594; Killingley et al. *J Infect Public Health* 2016; 9(3):278-288) than by infected ferrets.

It is true, both in guinea pigs and ferrets, and in other animal models, that some influenza virus strains transmit efficiently by an airborne route, while others don't, for reasons that are not always clear. The reviewer refers to Linster et al., which identified a minimal set of amino acid mutations in the viral proteins of an influenza A(H5N1) virus, which allowed airborne transmissibility in ferrets. They also very thoroughly characterized how those mutations altered the basic virology of the virus. Some of the mutations were, as they noted, "found in all pandemic influenza viruses of the last century and were therefore postulated to represent minimal requirements for adaptation of animal influenza viruses to humans to yield pandemic strains." Those mutations, which enable efficient polymerase function at lower mammalian temperatures and which increase virus binding to mammalian rather than avian sialic acid receptors, appear to be specific adaptations required for avian-to-mammalian host switching. However, Linster et al. (and others – e.g., Imai et al., *Nature* 2012; 486:420-428) found that substitutions that enhance the thermostability of the HA protein or decrease the pH at which its fusion activity is triggered are also favored in ferret transmission. Importantly, Linster et al. noted that, although the ability of these HA mutations "to increase airborne transmission between ferrets may be related to the pH of fusion or thermostability, these properties may merely be a surrogate for another—as yet unknown—phenotype, such as stability of HA in aerosols, resistance to drought, stability in mucus, or altered pH in the host environment." We emphasize this important observation: while some genetic adaptations appear to be required solely to allow the virus to replicate in a new host, some (like environmental stability) may confer fitness advantages outside of the host, and some may even result in multiple beneficial phenotypes.

Indeed, Pulit-Penalzo et al. observed in ferrets that "influenza viruses known to transmit efficiently through the air display enhanced stability in an aerosol state for prolonged periods compared to those viruses that do not transmit as efficiently" (*Appl Environ Microbiol* 2019; 85(10):e00210-19). Singanayagam et al. (*PLoS Pathog* 2020; 16(2): e1008362) published a novel method for capturing viable influenza viruses emitted by infected ferrets, which they call respiratory droplets. (Importantly, however, nothing in their apparatus precludes aerosolized fomites from the ferret's body from being detected by their method.) Of relevance to this discussion, they generated a recombinant mutant of a 2009 pandemic H1N1 influenza isolate with an HA mutation that alters its pH stability, "increasing the pH of fusion to 5.9, a level similar to highly pathogenic avian H5N1 viruses that do not transmit via the airborne route." They found that this virus, "containing a mutation destabilising the haemagglutinin (HA) surface protein displayed reduced survival in air," just as Linster et al. had hypothesized.

Our goal is not to challenge the validity of the ferret model. Importantly, past data derived in animal models about airborne influenza virus transmission are not necessarily invalidated

by or incompatible with the existence of aerosolized fomites. Similar decay mechanisms may act on influenza viruses in respiratory secretions that are evaporating while suspended in the air or after deposition on aerosolizable substrates. We are, however, suggesting that influenza virus transmission via aerosolized fomites in animal experiments may be an “unknown unknown” – something that we don't know we don't know. To interpret animal data meaningfully, we must understand how the animal does or does not model humans. A main goal of our paper is to show that aerosolized fomites do demonstrably exist; whether they do or do not play a role in influenza virus transmission in other species, including humans, remains to be investigated.

To address this point, we have added an abbreviated version of this discussion to the manuscript (lines 227-258), and we have emphasized that aerosolized fomites should be considered in all animal experiments, not just those with ferrets (lines 219-222).

In Fig 2A, LOD lines are misplaced (at least in my copy).

They appear correctly placed in our figures. Please note that there are two LODs, lower and upper. We have added both LODs (in pfu/ml of swab eluate) to the caption for clarity.

In L227-233, the authors highlight the background particle counts that are well known in the field and describe a “wash out” that is commonly used to deal with this problem in aerosol science. Was this wash out only done here, and not in other experiments?

The wash-out was performed in all experiments involving anesthetized and euthanized animals, as well as our tissue rubbing experiments, to minimize the influence of any residual particles. With the awake, mobile animals, a wash-out was performed prior to placement of the animal, but it meant little in practice because the animals immediately start moving around and aerosolizing particles the moment they are placed in the cage. We have revised the methods section to specify when wash-outs were performed.

Reviewer #2

We thank the reviewer for the positive assessment of our work and for the suggestions, both of which we have implemented. Detailed responses are listed below.

A. Summary of the key results 1. Virus-contaminated resuspended particles can transmit influenza in the guinea pig model.

B. Originality and significance. This work is highly original and significant because it shows that a previously underappreciated route of transmission—

deposition of virus onto surfaces followed by resuspension and subsequent exposure to these—is possible.

C. Data, methodology, statistics. The methods are valid.

D. Conclusions. The conclusions are justified. The researchers showed, through multiple lines of evidence, that virus-containing particles are resuspended by physical motion and that exposure to them can initiate infection.

E. Clarity and context. The writing, figures, and overall presentation are very clear. The authors cite prior work appropriately to place this study in context.

1. line 45: “Uncertainty surrounding the modes by which influenza virus transmits among humans under different conditions hinders the development of interventions, like vaccines, designed to prevent influenza’s spread.” I can see how uncertainty hinders the development of non-pharmaceutical interventions, but the link to vaccine development is not obvious to this reader.

We thank the reviewer for pointing out this potential point of confusion. We meant that animal models are widely used to test the efficacy of vaccines, and that more attention is being paid to prevention of transmission in the characterization of new influenza vaccine constructs (e.g., McMahon et al., mBio. 2019;10(3): e00560-19). But we agree that our original wording was too subtle. We have revised that sentence to explicitly refer to non-pharmaceutical interventions, which we agree is more directly connected to our results, and we have also modified the wording to clarify the connection to vaccine development.

2. Figure 4: Images of the plaque assay plates are not needed.

We thank the reviewer for this suggestion. We have moved the images to the supplementary material, and used the space saved in the main manuscript to address the points raised by reviewer 1.

REVIEWER COMMENTS, second round:

Reviewer #1 (Remarks to the Author):

The authors have done an excellent job revising the manuscript. The only thing I would recommend additionally is to replace "animal model" in the abstract on two occasions to "guinea pig model". The contribution of the newly identified route of transmission has yet to be demonstrated in mice, hamsters, pigs, ferrets and other animal models.

Response to Reviewers of “Influenza A is Transmissible via Aerosolized Fomites,” by Asadi et al. (NCOMMS-20-08163A)

Reviewer #1:

The authors have done an excellent job revising the manuscript. The only thing I would recommend additionally is to replace "animal model" in the abstract on two occasions to "guinea pig model". The contribution of the newly identified route of transmission has yet to be demonstrated in mice, hamsters, pigs, ferrets and other animal models.

We thank the referee for raising this point. We modified the abstract as requested for the first instance of the phrase “animal model” to clarify that our work focused on the guinea pig model. We did, however, retain the use of the word “animal” later in that same sentence for variety, to avoid repeating “guinea pig” twice in the same sentence:

In the **guinea pig** model of influenza virus transmission, we show that the airborne particulates produced by infected animals are mainly non-respiratory in origin.

We also changed “animal” to “guinea pig” in the next sentence, to further emphasize that our experiments were performed with guinea pigs:

Surprisingly, we find that an uninfected, virus-immune **guinea pig** whose body is contaminated with influenza virus can transmit the virus through the air to a susceptible partner in a separate cage.

In the last sentence of the abstract, we retained the phrase “animal models” because we are referring here to the broader implication of our results:

Our data suggest that aerosolized fomites may contribute to influenza virus transmission in animal models of human influenza, if not among humans themselves, with important but understudied implications for public health.

As discussed at length in the manuscript, our results demonstrate the biological plausibility of transmission via aerosolized fomites, although the phenomenon remains to be experimentally investigated in other animals, including humans. We believe it would be misleading to imply here that our results potentially apply only to guinea pigs and humans; whether it occurs in other animal models of influenza virus transmission remains an open and unexplored question. Accordingly, we think it appropriate to retain this instance of “animal models,” particularly as we are employing the modal verb “may” to indicate possibility rather than certainty.